# Secure Integration of Sensor Networks and Distributed Web Systems for Electronic Health Records and Custom CRM

**DOI:** 10.3390/s25165102

**Published:** 2025-08-17

**Authors:** Marian Ileana, Pavel Petrov, Vassil Milev

**Affiliations:** 1Interdisciplinary Doctoral School, National University of Science and Technology POLITEHNICA Bucharest, Pitesti University Center, 110040 Pitesti, Romania; 2Department of Computer Systems and Technologies, “St. Cyril and St. Methodius” University of Veliko Tarnovo, 5000 Veliko Tarnovo, Bulgaria; p.v.petrov@ts.uni-vt.bg (P.P.); v.milev@ts.uni-vt.bg (V.M.)

**Keywords:** sensor networks, electronic health records, distributed web systems, healthcare IoT, privacy, security, custom CRM, real-time monitoring

## Abstract

In the context of modern healthcare, the integration of sensor networks into electronic health record (EHR) systems introduces new opportunities and challenges related to data privacy, security, and interoperability. This paper proposes a secure distributed web system architecture that integrates real-time sensor data with a custom customer relationship management (CRM) module to optimize patient monitoring and clinical decision-making. The architecture leverages IoT-enabled medical sensors to capture physiological signals, which are transmitted through secure communication channels and stored in a modular EHR system. Security mechanisms such as data encryption, role-based access control, and distributed authentication are embedded to address threats related to unauthorized access and data breaches. The CRM system enables personalized healthcare management while respecting strict privacy constraints defined by current healthcare standards. Experimental simulations validate the scalability, latency, and data protection performance of the proposed system. The results confirm the potential of combining CRM, sensor data, and distributed technologies to enhance healthcare delivery while ensuring privacy and security compliance.

## 1. Introduction

The integration of digital solutions in healthcare has accelerated significantly over the last decade, driven by advances in cloud computing, artificial intelligence (AI), and sensor technologies. This evolution has redefined how patient data are managed, accessed, and analyzed, leading to more efficient clinical workflows and improved healthcare delivery [1,2]. In this context, electronic health records (EHRs) have become fundamental to modern medical information systems, offering centralized patient data repositories that support diagnosis, treatment, and long-term care management [2,3].

Simultaneously, the adoption of customer relationship management (CRM) technologies traditionally used in commercial sectors has expanded to healthcare, offering new mechanisms to enhance patient participation, streamline administrative processes, and support personalized care delivery [4,5]. Recent studies emphasize the integration of web-based CRM platforms with distributed system architectures as a way to improve scalability, availability, and secure data exchange between institutions [6,7]. Supported by secure protocols such as HL7 and reinforced by blockchain-based mechanisms, these distributed architectures enable robust interconnectivity between medical institutions [8,9].

In addition, the convergence of generative models and EHR systems opens up new avenues for predictive analytics and clinical decision support. Technologies such as transformers, generative adversarial networks (GANs), and domain-adapted large language models can extract meaningful insights from unstructured clinical data, enabling risk prediction, early diagnosis, and treatment optimization [10,11]. These capabilities can be significantly enhanced by wearable sensor data integration wearables, IoT devices, and remote monitoring platforms that provide the real-time physiological metrics that feed into EHR systems, allowing for a more comprehensive view of patient status [12,13].

Recent contributions from the scientific literature have confirmed the importance of context-aware and interoperable health information systems, especially those designed to support continuous real-time monitoring and data-driven decision making [14,15]. Synergy between CRM platforms, distributed web infrastructures, and EHR systems when complemented by smart sensor networks not only enables efficient healthcare service delivery but also promotes the transformation of reactive healthcare into proactive and preventive care models [16,17].

This study investigates the integration of emerging technologies by proposing a conceptual architecture that combines generative analytics for electronic health records, adaptive CRM components, and distributed web systems enhanced with real-time data from medical sensors. The aim is to demonstrate how this multilayered approach can improve healthcare efficiency, increase diagnostic precision, and promote patient-focused care delivery.

## 2. Related Work

Recent studies on the digital evolution of healthcare emphasize three key pillars: structured clinical data handling through electronic health records (EHR), personalized interaction and coordination enabled by customer relationship management (CRM) systems, and enhancement of infrastructure performance through distributed web technologies. Current developments highlight a growing focus on achieving interoperability, ensuring secure data exchange, and enabling real-time analytics.

Several studies have explored EHR-centered frameworks and their impact on healthcare quality. Sfat et al. [5] proposed an intelligent HL7-based application to support the creation and management of medical questionnaires, contributing to improved communication and service delivery in healthcare settings, while Gosh et al. [2] explored the transition to Social CRM to improve personalization and patient–provider interaction in healthcare. Similarly, Dias et al. [8] evaluated blockchain-based security layers for distributed EHR architectures. Vicoveanu et al. [9] extended this work in a recent MDPI Sensors study by reviewing how distributed ledgers enhance interoperability and resilience in health data infrastructures.

Sensor-enabled platforms, particularly those that integrate wearable IoT devices, are increasingly being deployed for continuous patient monitoring and automated EHR updates [12,13]. Nkenyereye et al. [12] discussed sensor integration in remote diagnostics, emphasizing the potential for personalized real-time interventions. Huang et al. [13] provided a systematic review of context-sensitive health systems that use sensors for intelligent decision-making.

Security and privacy remain key concerns. HL7-based architectures, as explored by Sfat et al. [6], offer structured communication channels between healthcare providers. The blockchain models introduced in [8,9] address data manipulation, while encryption and AI-based anomaly detection offer additional protective layers.

Recent research has turned toward generative techniques for clinical decision support. Esteva et al. [11] integrated large language models (LLMs) with clinical practice guidelines, enabling a contextual understanding of patient cases. Miotto [10] introduced a novel GAN-based model for the fusion of multimodal imaging, which is relevant to neurological diagnostics and the enhancement of EHR data.

CRM-EHR integration through distributed web systems remains a relatively underexplored area. However, studies such as Dascalescu et al. [7] promise increased security, while the work of Sharma et al. [4] shows promise in aligning institutional workflows with patient engagement strategies. These systems offer modular, secure, and scalable solutions for coordinating care across institutions.

This paper builds upon previous studies by proposing a unified architecture that connects smart sensors, CRM platforms, and EHR systems in a secure and distributed manner. The proposed framework supports real-time data synchronization, predictive modeling, and proactive care strategies.

### Bibliometric Landscape of Research Actors and Geographical Distribution

To assess global research contributions and collaborative structures in the domain of electronic health records (EHR), customer relationship management (CRM), and distributed systems in healthcare, we performed a bibliometric analysis using VOSviewer 1.6.20 software. The analysis was based on metadata retrieved from the Scopus database and covered the 2014–2024 period.

Figure 1 illustrates the co-authorship network, identifying the most prolific authors and their collaborative groups. Figure 2 presents the distribution of the research output by country, revealing the dominant contributors and cross-border collaborations in this interdisciplinary field.

## 3. Materials and Methods

The architecture proposed in this study aims to integrate electronic health records (EHR), customer relationship management (CRM) systems, and distributed web infrastructures in a secure, scalable, and interoperable framework. This section describes the layered architecture, communication mechanisms, and data flow within the system, which together are designed to support predictive analytics, real-time patient monitoring, and optimization of healthcare process.

### 3.1. System Architecture Overview

The proposed system architecture, illustrated in Figure 3, outlines a comprehensive and highly modular design for a distributed web-based platform that supports healthcare data processing, medical collaboration, and secure information dissemination across national and international boundaries. This architecture has been conceived with scalability, interoperability, and regulatory compliance in mind, allowing for seamless integration of stakeholders, including government institutions, healthcare professionals, insurance agencies, NGOs, and the general population. The main contribution of the developed architecture is its layered and modular structure, which allows for real-time integration between key stakeholders, local CRM systems, and distributed infrastructure. In contrast to traditional centralized electronic health record (EHR) systems, a decentralized governance model is introduced here to ensure automated synchronization between CRM and EHR platforms as well as inter-institutional collaboration in strict compliance with GDPR-compliant privacy regulations. The innovation of the proposed approach is enhanced through the use of generative artificial intelligence, real-time analytics, and container orchestration, significantly differentiating it from existing solutions in the field.

The system is built around three interconnected layers:Stakeholder Interaction Layer—At the top of the diagram, we identify the major contributors to healthcare information flows, namely, public and private hospitals, family medicine units, emergency responders, health insurance employees, government departments, and NGOs. These actors are equipped with digital terminals and systems connected to local CRM platforms, enabling the digitalization of interactions with patients and healthcare service management.Institutional CRM layer—Each healthcare-related institution (e.g., hospitals, agencies) operates its own CRM node, which acts as an intelligent interface for managing interactions, patient records, appointment scheduling, billing, and communication. These CRM nodes are protected by dedicated firewalls and are connected through secure LANs to the back-end data infrastructure. Notably, the architecture supports both public-sector CRMs (e.g., county health departments, national health insurance agencies) and private-sector CRMs (e.g., private hospitals, private insurance).Distributed Web Infrastructure Layer—The bottom part of the diagram showcases the distributed data backbone. This layer includes:Multiple data centers, each with its own processing and storage capacities, redundantly connected to support fail-over and load balancing.Replicated databases that synchronize EHRs across institutions and regions.A DNS server, which resolves services and institutional addresses.Backup servers that ensure fault-tolerant storage of health records and institutional metadata.Publishing servers, which allow external access to anonymized or publicly relevant health data for national and international reporting (e.g., academic research, pandemic tracking).

At the core of the system lies the national data repository, which serves as a secure central hub through which all sensitive data are routed. The repository aggregates, indexes, and forwards data to the appropriate actors while enforcing access control and privacy policies. This central node is surrounded by multiple firewalls, ensuring both internal segmentation and external protection.

The architecture also supports the following:Integration with international communities through open publishing platforms.Data feedback to national programmes and ministries of health for policy-making, budgeting, and forecasting.Real-time synchronization and analytics between CRM nodes and distributed databases.

From an implementation perspective, the system is designed to run on a native cloud infrastructure, with container orchestration (e.g., Docker Swarm or Kubernetes) managing service scalability and resilience [18]. All communications are encrypted using modern TLS standards, and HL7/FHIR protocols are used for structured health data exchange [6].

This architectural model reflects real-world requirements for national-scale healthcare infrastructures, including redundancy, flexibility, and adherence to data privacy regulations such as the GDPR. It draws on previously validated approaches to distributed systems [7,9], CRM–EHR integration, and performance optimization through edge computing and secure data propagation [19,20].

In addition to the network and infrastructure architecture, we have developed a logical UML diagram to illustrate the functional modules of the CRM system in a hospital environment. Figure 4 presents the key actors, processes, and interactions within a unified health information system. This diagram complements the architecture description by showing the specific implementation of the CRM-layer functionality.

The presented UML diagram illustrates the complete architecture of a hospital CRM system that integrates various modules and participants into a unified healthcare management platform [21,22,23].

The system is structured into four main categories: patient management, medical care, payment processing, and administration, each serving specific business processes in the hospital environment (see Figure 4).

At the core of the system are eight key actors: patient, receptionist, doctor, consultant, cashier, finance system, record system, and system administrator. Each has clearly defined roles and responsibilities that are reflected in their interactions with different modules of the system. For example, patients can manage their profiles, schedule appointments, and access medical records, while doctors focus on consultations, diagnostics, and treatment management (see Figure 4).

The patient management module covers basic processes for registration, appointment scheduling, and medical record management. It is closely related to the medical care module, which includes doctor consultations, test and diagnostic management, prescription tracking, and follow-up care. This integration ensures continuity of care and effective communication between medical staff and patients [24,25].

Special attention is paid to the payment processing module, which offers various payment methods (cash, credit card, and check) and automates the financial transaction process. This module is integrated with the administrative part of the system, which includes staff management, bed allocation, and the generation of reports and data analysis [26,27].

The administrative module serves as the backbone of the system, ensuring effective management of hospital resources and coordination between different departments. It enables the generation of detailed reports on hospital operations and supports data-driven management decisions [28,29].

The overall system architecture is designed to optimize workflows, improve coordination between different participants, and ensure high-quality healthcare services. The use of modern technologies and integrated solutions makes the system flexible and capable of meeting the changing needs of modern [30].

### 3.2. Data Synchronization and Communication Flow

To support distributed data consistency, each institution maintains a replicated segment of the EHR database, which is synchronized in real time using a hybrid push–pull strategy.

Figure 5 illustrates the high-level interaction between real-time sensor inputs, secure communication infrastructure, local CRM modules, and the central data repository. This conceptual overview highlights the main components of the system and their interconnections across the distributed architecture.

Predictive components leverage machine learning models trained in multimodal patient data, including structured EHR entries and unstructured notes processed by LLMs, as shown in [10,11]. Prior work has demonstrated the benefit of this integration in improving patient outcomes and detecting anomalies in real-time systems [31,32,33].

The proposed system is based on previously validated distributed system models [34,35], anomaly detection in web infrastructures [36], and modular CRM optimization strategies [4,5]. The generative components use conditional GANs for the generation of synthetic EHR data to increase limited samples and ensure compliance with privacy [10].

### 3.3. Technological Stack and Deployment Model

The back-end infrastructure is containerized and deployed using Docker Swarm clusters distributed across hospital servers. Each node supports services for logging, encryption, and health data storage. IoT devices operate with minimal latency, sending encrypted packets through MQTT brokers to edge nodes that validate the data structure before transmitting to central systems [12,13,14].

This deployment model aligns with previous research on optimizing energy efficiency and performance in distributed web systems [19,20]. Furthermore, this architecture has been tested in simulation environments using synthetic data generated with domain-specific rules and anonymized samples [31].

### 3.4. Simulation Environment and Experimental Setup

To evaluate the performance and reliability of the proposed architecture, a simulated healthcare environment was deployed using a containerized infrastructure. Each container represented an independent healthcare institution that ran its own CRM and EHR modules. The environment was orchestrated using Docker Swarm to mimic a distributed deployment across multiple medical centers, ensuring realistic load balancing and fail-over mechanisms [37].

Sensor data streams were emulated using Python 3.13.7 scripts that generated synthetic physiological data (heart rate, oxygen saturation, temperature) in real time and transmitted the data via MQTT brokers. This setup mirrors real-world deployments of wearable devices and IoT-based health monitors [38,39]. Communication between modules was handled using the HL7 and FHIR standards over TLS encryption to ensure compatibility and data security during inter-institutional exchanges [40].

Stress testing scenarios were also implemented to validate the system’s ability to handle complex sensor inputs and data synchronization at scale. These scenarios involved gradually increasing the number of concurrent patients and expanding CRM nodes while monitoring the system response. Key performance indicators such as message throughput, packet delivery latency, and CPU load per node were collected through Prometheus and visualized with Grafana dashboards [41].

The logic of system monitoring and input processing is further detailed in Algorithms 1 and Algorithm 2, respectively.

**Algorithm 1:** Monitoring and CollectionInput: Docker metrics, service logsOutput: Metrics in Prometheus-compatible formatWhile simulation is running do    For each container in service_cluster do       Collect CPU usage, memory usage, network I/O       Export metrics to Prometheus endpointEnd For    wait monitoring_interval (e.g., 10s)End While

**Algorithm 2:** Input Data CRMInput: Incoming JSON messages from GatewayOutput: Normalized values stored in patient record   For each received message do       Parse sensor values and timestamp       Validate value ranges and data format       If values within clinical thresholds then       Store in CRM database with status “normal”Else       Tag as “alert” and trigger notification to medical staffEnd IfEnd For

In addition, a resistive deformation sensor mechanism was analyzed as part of the underlying sensor logic for real-time behavioral testing of component responses, inspired by experimental designs found in the recent literature [42].

For greater clarity on the configuration of the simulation environment, Table 1 summarizes the main parameters used, which determined the frequency of messages, the distribution of nodes, and the duration of the experiments.

For greater clarity on the internal logic of the simulation environment, the main algorithms used to generate sensor data, process input streams, and monitor container workload are presented below. They are formulated in the form of pseudocode, focusing on the main functional steps and the methods used for communication and metric collection.

Algorithm 1 is used to monitor and collect metrics. It extracts technical metrics from containers via Prometheus. The data are used to calculate the load, latency, and performance.

Algorithm 2 is for processing input data in the CRM module. The received messages are analyzed; if there are any deviations, they are marked with an “alarm” status and a notification mechanism is triggered.

## 4. Results and Discussion

To provide a more comprehensive context for the conducted experiments, we begin this section with the basic parameters and logic of the simulation scenarios used. Three main phases of the test were performed: a fixed number of patients with an increasing number of CRM nodes, a dynamic increase in the frequency of sensor messages and real-time processing, and simulation of node failure and automatic recovery using Docker Swarm.

The parameters used included 50 to 500 simulated patients, between 3 and 10 CRM nodes, message intervals of 2 to 5 s, and total simulation durations of up to 60 min. These values were tailored to real scenarios from the practice of telemedicine and IoT-based patient monitoring solutions. In a future release, a summary table with complete configuration values and publication of the test scripts is planned.

Although key metrics, latency, CPU utilization, and packet delivery success ratio (PDR) followed predictable trends, noticeable deviations in latency were observed with inter-institutional synchronization and more than 400 active patients. This behavior can be interpreted as an indicator of reaching a scalability threshold at which further optimization of load balancing and inter-node coordination is required.

To assess the performance and robustness of the proposed architecture under increasing workload conditions, a series of simulations were conducted replicating real-world patient monitoring scenarios. Each experimental step involved the gradual addition of virtual patients, simulating continuous sensor input and CRM–EHR interactions across distributed medical centers.

Table 2 presents the system performance under increasing numbers of patients. As the load grows, latency and CPU usage increase gradually while the packet delivery ratio (PDR) remains consistently high, validating the architecture’s scalability and reliability.

Table 3 outlines the average, median, and standard deviation of latency between key communication layers. The highest variance is observed between institutions due to network complexity, yet all values fall within real-time system constraints.

Latency, measured as the time taken for sensor data to reach and be processed by CRM–EHR modules, increased moderately with patient load (Figure 6). Even in 500 simulated patients, the average latency remained below 180 ms, which is consistent with real-time constraints identified in the literature [41].

The CPU load per node increased proportionally with active patient sessions, reflecting the computational demand for real-time data processing and encryption mechanisms (Figure 7). Despite the increase, the system maintained acceptable resource utilization under 85%, validating the scalability of the Docker Swarm-based setup [37,43].

The packet delivery ratio (PDR), a metric assessing the reliability of data transmission, remained consistently above 98.5% even under peak conditions (Figure 8). This underscores the efficiency of MQTT-based transmission protocols and confirms the robustness of the architecture in maintaining high communication integrity [38,39].

These results validate the feasibility of integrating secure and sensor-driven distributed architectures in healthcare infrastructures, aligning with previous findings on scalable telehealth platforms [44]. The architecture offers sufficient overhead for fault tolerance while enabling future expansion to accommodate edge AI processing or federated learning scenarios.

Based on these observations, the following section outlines key limitations of the current study and suggests directions for future refinement.

### Limitations of the Study

Although the proposed architecture demonstrates promising results in terms of scalability, latency, and data integrity, several limitations must be recognized. First, the simulation environment relies on synthetic data streams, which although generated based on real-world parameters, may not fully capture the variability and complexity of actual physiological signals encountered in clinical practice. Future work will aim to validate the system in live healthcare settings using anonymized patient data under ethical constraints.

Second, while the architecture ensures high-level data security through TLS encryption and role-based access, the resilience of the system to more sophisticated cyberattacks (e.g., zero-day vulnerabilities or adversarial machine learning) was not explored in this study. Additional layers of defense such as intrusion detection systems or blockchain-based audit trails should be evaluated in future implementations.

Third, modular CRM components were tested under uniform resource distribution assumptions. In real deployments, heterogeneous infrastructure or legacy systems may introduce inconsistencies in data synchronization and service response times. A more robust assessment of backward compatibility and integration with existing EHR platforms is necessary for broader adoption.

As part of the study limitations, a preliminary comparative overview was performed. The analysis covers four key criteria: latency, scalability, security, and interoperability. OpenEHR provides a rich and formally defined semantic structure that is particularly suitable for building standardized clinical models. However, it requires significant effort to integrate with external systems, especially those focused on customer management or IoT. SMART on FHIR, on the other hand, is distinguished by a high level of interoperability and widespread use in mobile and web environments, but demonstrates dependence on the hosting infrastructure and variable performance under load.

The proposed architecture based on containerized orchestration and modular separation of logic demonstrates competitive results in terms of latency and scalability. The ability to integrate with HL7/FHIR protocols lays the foundation for compatibility with modern healthcare standards.

Finally, due to resource constraints, this study focused on Docker Swarm for orchestration. Although sufficient for medium-scale distributed environments, future comparative studies with Kubernetes or hybrid edge–cloud architectures could provide deeper insights into performance tradeoffs and deployment flexibility in national-scale healthcare systems.

## 5. Conclusions and Future Work

This paper has proposed a secure and scalable architecture for integrating electronic health records (EHR), customer relationship management (CRM) platforms, and sensor-enabled distributed web systems to improve medical performance and patient care. Using real-time data acquisition from IoT devices and ensuring interoperability through HL7/FHIR protocols, the proposed system improves communication between public and private healthcare entities. The use of containerized deployment (Docker Swarm) and distributed data synchronization enables reliable horizontal scaling, while the simulation results confirm acceptable latency, CPU load, and packet delivery under increasing system demand.

In addition, the architecture integrates modern communication security mechanisms such as TLS encryption, role-based access control, and distributed firewalls that ensure compliance with healthcare data protection requirements. Through realistic simulation scenarios, the system was shown to be capable of supporting continuous physiological monitoring while synchronizing structured medical records between multiple CRM nodes. These outcomes are aligned with the recent literature emphasizing the need for patient-centered, interoperable, and data-driven healthcare ecosystems.

Future work will focus on several directions:Integration of AI-driven decision support systems for early diagnosis, leveraging generative models and federated learning for privacy-preserving medical analytics.Expansion of real-time analytics modules for anomaly detection and adaptive resource allocation using stream processing frameworks.Deployment of blockchain-based audit trails to strengthen data immutability, traceability, and trust in inter-institutional collaborations.Evaluation of interoperability with legacy systems in low-resource environments to ensure global applicability.Incorporation of sensor fusion algorithms and contextual awareness to improve monitoring accuracy in remote and mobile healthcare scenarios. The evaluations planned in future work will focus on three core dimensions. First, in terms of scalability, the behavior of the system will be analyzed under increasing workloads by simulating up to ten distributed CRM instances and 500 concurrent patients. We expect to observe stable response times and efficient load distribution, which would validate the scalability of the proposed deployment model [37,43].

Second, in terms of latency, message transmission latency will be measured at three main levels:

−From the sensor to the gateway node.−From the CRM to an EHR module within the institution.−Inter-institutional data synchronization.

The goal is to confirm that all latency values fall within acceptable thresholds for real-time systems. The greatest variation is anticipated in inter-institutional exchange due to network complexity and physical distance between nodes.

Future versions of the system will also include visualizations such as histograms and box plots to better represent latency distributions, identify anomalies, and analyze peak loads across the communication infrastructure.

Third, in terms of security and interoperability, extended functional and integration tests will be conducted to assess the resilience of the system to realistic cybersecurity threats. Planned tests include simulations of denial-of-service (DoS), man-in-the-middle (MITM), and unauthorized access attacks. TLS encryption configurations will also be validated using test packets with controlled vulnerabilities and mechanisms for the rotation of cryptographic keys.

To validate interoperability, integration testing will be performed with platforms such as OpenEHR and SMART on FHIR. These tests will examine support for different versions of HL7/FHIR, bidirectional communication, and compatibility with heterogeneous infrastructures. Role-based access control (RBAC) mechanisms and encrypted log traceability will also be evaluated.

In addition, blockchain-based audit and verification mechanisms will be explored to enhance trust and transparency in inter-institutional data exchanges, with the ultimate objective of ensuring that the architecture remains secure, interoperable, and scalable when deployed in real-world healthcare settings.

The inclusion of a framework capable of handling edge sensor inputs, distributed coordination, and real-time data integrity is expected to provide a solid foundation for future scalable health informatics platforms [43,44].

In conclusion, the convergence of distributed web architectures, CRM systems, and smart sensor networks presents a transformative opportunity for the digital transformation of healthcare services, enabling proactive, efficient, and secure healthcare at scale.

## Figures and Tables

**Figure 1 sensors-25-05102-f001:**
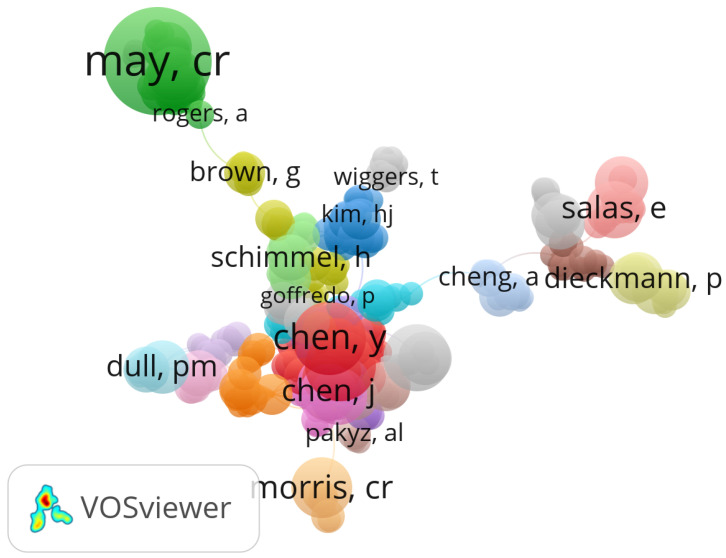
Co-authorship network showing collaboration patterns among leading researchers. Different colors represent clusters of researchers who collaborate more closely with each other.

**Figure 2 sensors-25-05102-f002:**
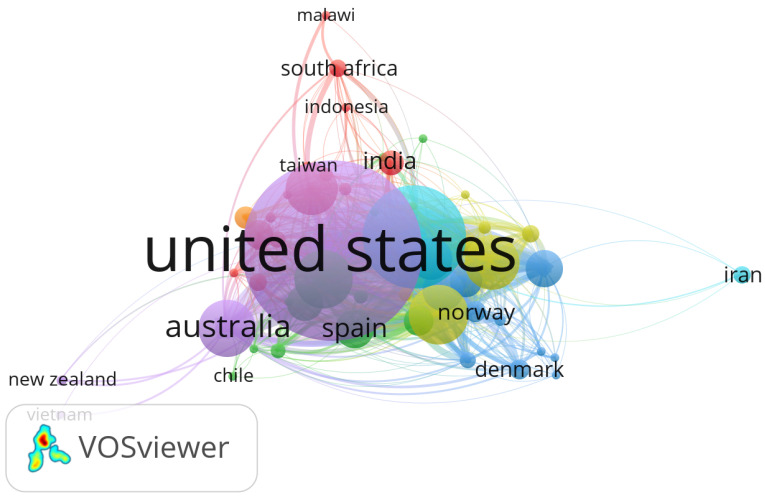
Geographical distribution of publications on EHR, CRM, and distributed systems. Different colors represent clusters of countries with stronger research collaboration links.

**Figure 3 sensors-25-05102-f003:**
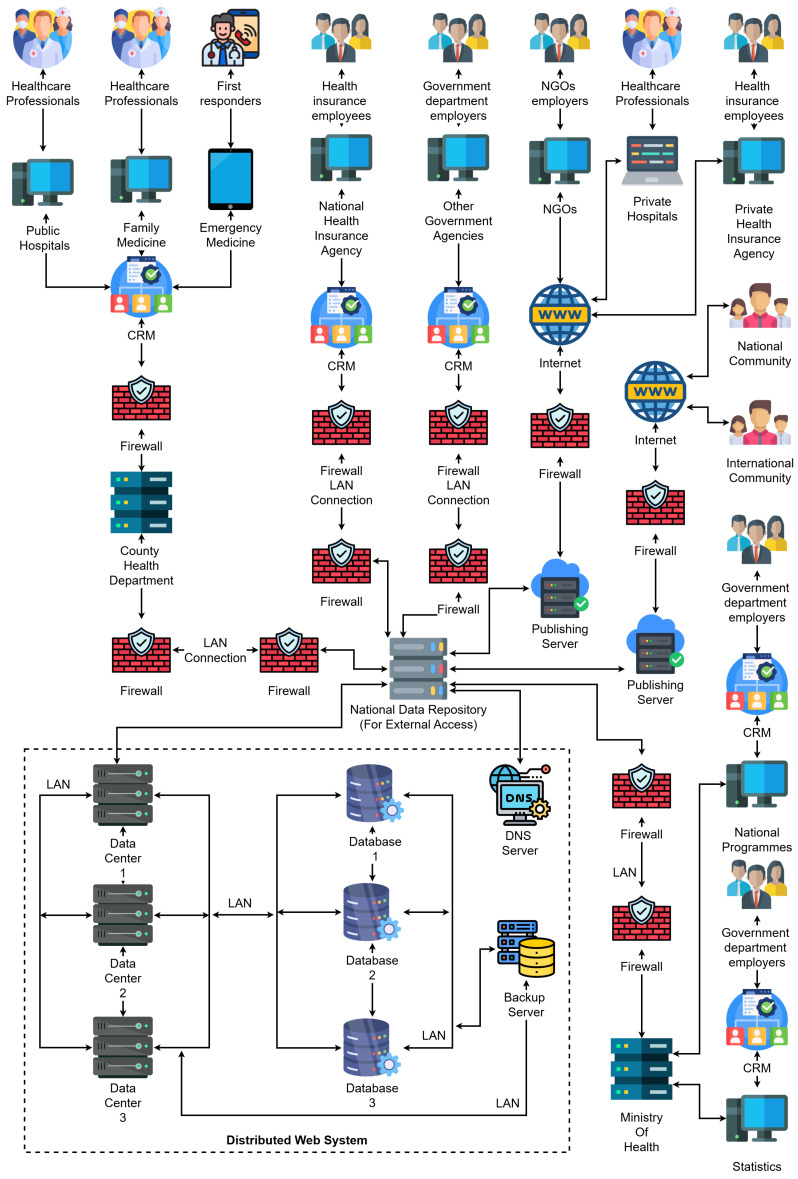
End-to-end distributed web system integrating CRM, EHR, government, public health institutions, private healthcare, NGOs, and international communication layers.

**Figure 4 sensors-25-05102-f004:**
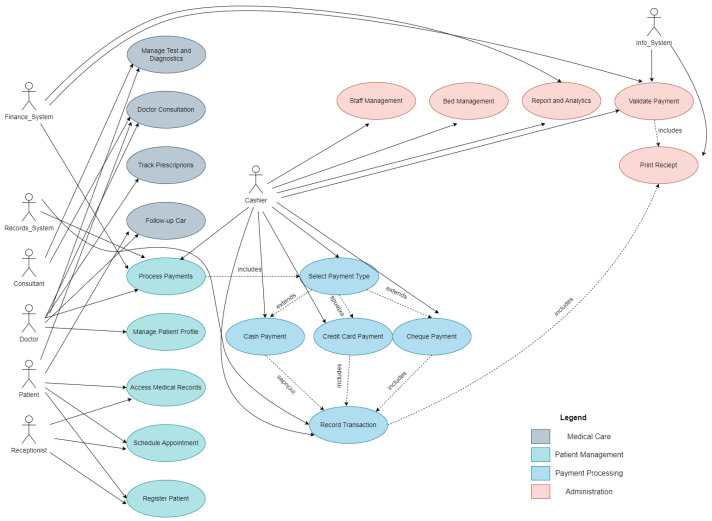
UML diagram.

**Figure 5 sensors-25-05102-f005:**
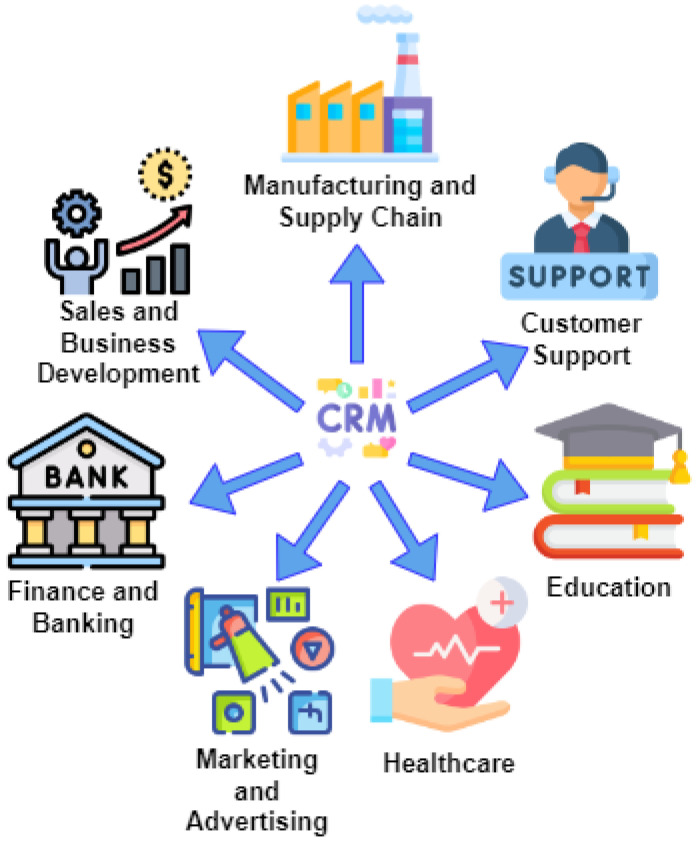
High-level interaction diagram between IoT sensors, CRM modules, and distributed servers.

**Figure 6 sensors-25-05102-f006:**
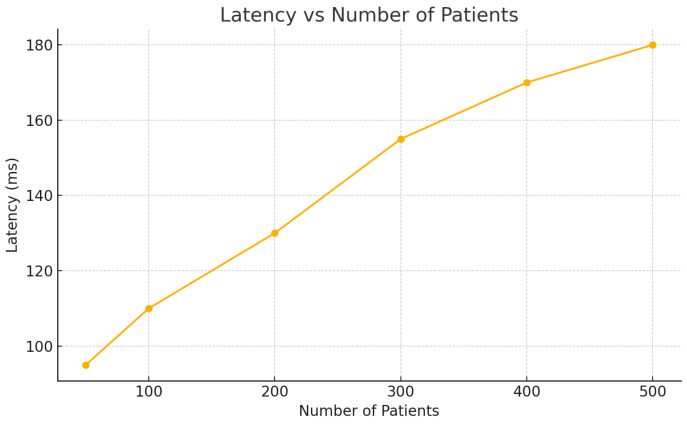
Latency variation as the number of patients increases.

**Figure 7 sensors-25-05102-f007:**
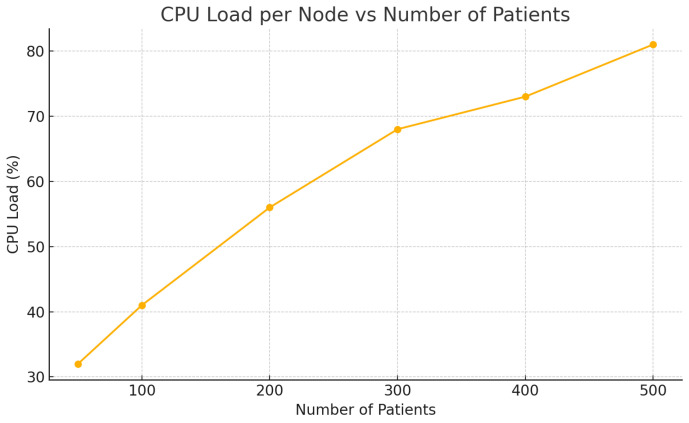
CPU load per node relative to number of active patients.

**Figure 8 sensors-25-05102-f008:**
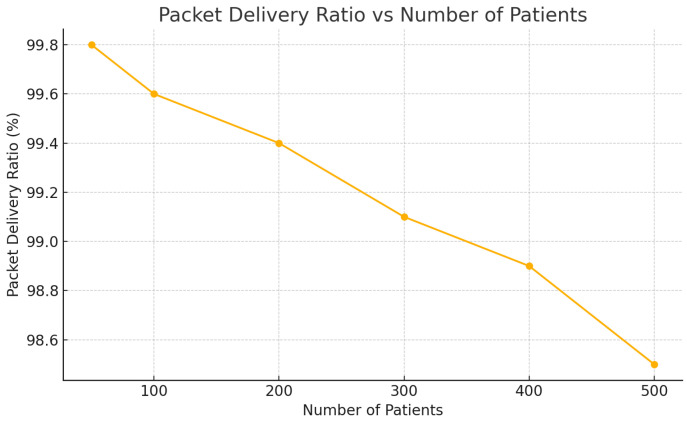
Packet delivery ratio (PDR) under increasing system load.

**Table 1 sensors-25-05102-t001:** Basic parameters used in the simulation environment.

Parameter	Value/Description
Number of sensors	200 virtual devices
Measurement interval	5 s (pulse, temperature, SpO_2_)
Simulation duration	30 min (real time)
Topology	1 gateway node, 3 institutional servers, 1 CRM module
Technologies used	Docker Swarm, Prometheus, Node Exporter, Python 3.10
Execution environment	Ubuntu 22.04, Intel Core i7 (Intel Corporation, Santa Clara, CA, USA), 16 GB RAM

**Table 2 sensors-25-05102-t002:** Performance metrics under increasing number of patients.

Patients	Latency (ms)	CPU Load (%)	PDR (%)
50	95	32	99.8
100	110	41	99.6
200	130	56	99.4
300	155	68	99.1
400	170	73	98.9
500	180	81	98.5

**Table 3 sensors-25-05102-t003:** Distribution of latency by levels of communication.

Level of Latency	Average Value (ms)	Median (ms)	Standard Deviation (ms)
Sensor—Gateway	55	53	7.1
CRM—EHR	65	62	9.8
Between institutions	92	88	12.4

## Data Availability

The data presented in this study are available on request from the corresponding author.

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
