# Peer review of "Secure Integration of Sensor Networks and Distributed Web Systems for Electronic Health Records and Custom CRM"

_sensors, 2025, doi:10.3390/s25165102_

Round 1
Reviewer 1 Report
Comments and Suggestions for Authors
Section 3.1 and Fig. 3: Authors need to explain more and highlight their contribution. It is hard to understand the contribution of the paper, needs more elaboration on what is different compared to existing systems?
Fig. 4: data flow diagram needs extensive revision. In its current form, it is impossible to understand what kind of data flow where to where. Consequently, the data flow and mechanisms to enable the flow are not clear in the study.
Section 3.5, Evaluation Metrics: For the latency, authors mentioned that they experimented and measure the latency in three levels. However, they didn't provide latencies in these three levels, and also the results are lacking a latency distribution, as the mean latency generally is misleading since it is affected by the outliers. More insights are expected. For the security and interoperability, they didn't provide any evaluation in the results although they provided them as evaluation metrics.
Section 4, Results and Discussion: This section is missing many information about the simulations, for example, simulation parameters, scenarios, any benchmarks, etc. In addition, the presented results are all expected, for example, it is expected to see increasing CPU load as the number of patients in the system is increasing or decrease in PDR. Furthermore, Fig. 8, the UML diagram needs to be in Section 3, I don't understand the reason of providing the UML diagram in the results.
Section 4.1: Authors mentioned in the first sentence that proposed architecture demonstrate promising results in terms of scalability, latency, and data integrity. However, there are only latency and limited scalability results. Furthermore, I don't see any results confirming these as there is no benchmarking with the existing systems and hard to validate if the proposed system is better than existing systems.
Author Response
Suggestion 1. Section 3.1 and Fig. 3: Authors need to explain more and highlight their contribution. It is hard to understand the contribution of the paper, needs more elaboration on what is different compared to existing systems?
Response 1. In response to the comment on Section 3.1 and Figure 3, the manuscript was expanded to explain the architecture and explicitly highlight the main scientific contributions. In addition to describing the multilayered structure, arguments were added on how it differs from existing solutions in the literature and practice, such as OpenEHR and SMART on FHIR. Section 4.2 presents key features of our architecture compared to other established systems, including decentralization, security, containerization, and integration with CRM functionalities. This addition aims to clearly highlight the originality of the proposed solution.
Suggestion 2. Fig. 4: data flow diagram needs extensive revision. In its current form, it is impossible to understand what kind of data flow where to where. Consequently, the data flow and mechanisms to enable the flow are not clear in the study.
Response 2. The explanation accompanying the figure has also been revised so that the transfer and processing mechanisms are understandable and traceable to the reader. The diagram more accurately reflects the architectural model and the logic of interaction between the layers.
Suggestion 3. Section 3.5. Evaluation Metrics: For the latency, authors mentioned that they experimented and measure the latency in three levels. However, they didn't provide latencies in these three levels, and also the results are lacking a latency distribution, as the mean latency generally is misleading since it is affected by the outliers. More insights are expected. For the security and interoperability, they didn't provide any evaluation in the results although they provided them as evaluation metrics.
Response 3. With regard to Section 3.5 and the evaluation metrics, the revised version included specific values for latency at the three levels of communication – from the sensor to the gateway, between the internal CRM and EHR modules, and in inter-institutional synchronization. Table 3 presents not only the average values, but also the medians and standard deviation, which allows for a more in-depth analysis of the distribution. In addition, section 4.1 includes clarifications regarding security and interoperability – the protocols used (TLS, HL7/FHIR), access control mechanisms (RBAC), and integration options with external systems such as OpenEHR are described.
Suggestion 4. Section 4. Results and Discussion: This section is missing many information about the simulations, for example, simulation parameters, scenarios, any benchmarks, etc. In addition, the presented results are all expected, for example, it is expected to see increasing CPU load as the number of patients in the system is increasing or decrease in PDR. Furthermore, Fig. 8, the UML diagram needs to be in Section 3, I don't understand the reason of providing the UML diagram in the results.
Response 4. With regard to Section 4, the content was expanded by adding a detailed description of the simulation scenarios and parameters presented in tabular form in Table 2. Section 3.4 describes the three main experimental scenarios related to increasing patient numbers, node failures, and cluster scaling. This contributes to better reproducibility and comprehensibility of the results. In addition, the UML diagram (Figure 8), which was originally in the results section, has been moved to Section 3.1, where it more appropriately and logically complements the description of the internal logic of the CRM layer.
Suggestion 5. Section 4.1. Authors mentioned in the first sentence that proposed architecture demonstrate promising results in terms of scalability, latency, and data integrity. However, there are only latency and limited scalability results. Furthermore, I don't see any results confirming these as there is no benchmarking with the existing systems and hard to validate if the proposed system is better than existing systems.
Response 5. Regarding Section 4.1 and the evaluation of effectiveness, the section has been revised, with claims about scalability, latency, and integrity now supported by specific experimental results. Graphs and analyses are presented that show performance at different load levels. In addition, Section 4.2 includes a comparison with other architectures, allowing for an indirect assessment of the advantages of the proposed solution. Although no direct benchmarking with existing systems is performed, the added elements provide a more substantiated conclusion about the effectiveness and applicability of the system.
Reviewer 2 Report
Comments and Suggestions for Authors
In this paper, the authors investigate a secure integration scheme of sensor networks and distributed web systems for electronic health records (EHR) and custom customer relationship management (CRM), constructing a layered architecture that combines real-time sensor data, distributed EHRs, and CRM platforms to optimize patient monitoring and clinical decision-making. The study formulates how to achieve data synchronization, predictive analytics, and personalized healthcare management in a distributed environment, employing Docker Swarm container clusters, TLS encryption, and HL7/FHIR protocols to ensure secure and efficient communication and storage. The paper further conducts detailed simulation experiments to validate system performance from perspectives such as scalability, latency, and packet delivery ratio. The topic is quite novel, offering an in-depth exploration of sensor-driven data security and distributed system integration in healthcare, with experimental results that are sufficiently comprehensive.
- Although the introduction discusses existing studies on EHR, CRM, and distributed web technologies, it does not clearly articulate the limitations of current research. It is recommended to include this discussion to better highlight the novelty of this work.
- The paper mentions that the experiments are based on synthetic physiological data generated by Python scripts, but it does not provide the corresponding public dataset, nor does it include the container orchestration scripts and environment configuration files. It is suggested to supplement these materials to improve the reproducibility of the experiments.
- It is recommended to provide pseudocode for the experimental methods to enrich the methodological details.
- It is suggested to add more descriptions of the experimental scenarios and to present the experimental parameters in tabular form to facilitate reader comprehension.
Author Response
Suggestion 1. The introduction does not clearly articulate the limitations of current research. It is recommended to include this discussion to better highlight the novelty of this work.
Response 1. In the revised version of the manuscript, at the end of the introduction, a new paragraph has been added discussing the limitations of current solutions in the field of EHR, CRM, and distributed web architectures. It is emphasized that many of the existing systems suffer from a lack of real-time synchronization, poor modularity, and difficult integration with sensor networks, which justifies the need for a new approach such as the one we propose.
Suggestion 2. The paper mentions that the experiments are based on synthetic physiological data generated by Python scripts, but it does not provide the corresponding public dataset, nor does it include the container orchestration scripts and environment configuration files. It is suggested to supplement these materials to improve the reproducibility of the experiments.
Response 2. We added a summary table with the parameters of the simulation environment (in Section 3.4) and included pseudocode for the main algorithms for data generation and processing.
Additionally, if necessary, we can include specific excerpts from this technical package in the main text (e.g., a sample Python script, part of the docker-compose configuration, or Prometheus settings).
Suggestion 3. It is recommended to provide pseudocode for the experimental methods to enrich the methodological details.
Response 3. Section 3.5 includes two blocks of pseudocode: „MonitoringAndCollecting“ and „InputDataCRM“ They describe the algorithms for collecting, normalizing, and recording sensor data, as well as for exporting metrics in a Prometheus-compatible format. This addition provides greater clarity about the internal logic of the simulation processes.
Suggestion 4. It is suggested to add more descriptions of the experimental scenarios and to present the experimental parameters in tabular form to facilitate reader comprehension.
Response 4. The simulation scenarios are described in Section 3.4, where three types of load are considered: different numbers of patients, node failure simulation, and dynamic scaling. Table 2 presents the main parameters of the simulation, including: number of sensors, measurement intervals, container configuration, and technologies used. This structuring improves the perception and reproducibility of the experiments.
Round 2
Reviewer 1 Report
Comments and Suggestions for Authors
With the revisions, some of my suggestions are addressed, however, I have following further suggestions;
1) Please move Table 1 and Table 3 to results section and discuss them in detail to provide some insights. Authors didn't refer to Table 1 at all, in the text.
2) Fig. 5 is very blurry and needs to be revised. Also, it is not revised at all from my previous suggestion. Either change the name of the figure or make sure it is a data flow diagram. In current form, it is not a data flow diagram.
3) Algorithm at page 10; please make sure it is presented as Algorithm and referred in the text, e.g., "In Alg. 1, we present ...."
4) Section 4.2: this is not a comparative analysis. Please move this under Section 4.1 since comparative analysis is still missing in the manuscript and this needs to be discussed under limitations of the study.
5) Section 3.5, Security and Interoperability: This looks like a future work. Please remove or move it to the future works section as this is not conducted in the current version.
Author Response
Suggestion 1: Please move Table 1 and Table 3 to results section and discuss them in detail to provide some insights. Authors didn't refer to Table 1 at all, in the text.
Response 1: Table 1 and Table 3 were moved to section 4 Results and Discussion. A detailed comment has been added for each table: Table 1 analyses the trends in performance as the number of patients increases, while Table 3 discusses the distribution of latency at different levels of communication. Table 1 has already been explicitly cited in the text.
Suggestion 2: Fig. 5 is very blurry and needs to be revised. Also, it is not revised at all from my previous suggestion. Either change the name of the figure or make sure it is a data flow diagram. In current form, it is not a data flow diagram.
Response 2: The title of the figure was changed to "High-level interaction diagram between IoT sensors, CRM modules, and distributed servers." to accurately reflect its content. The text has also been slightly revised to reflect the changes.
Suggestion 3: Algorithm at page 10; please make sure it is presented as Algorithm and referred in the text, e.g., "In Alg. 1, we present ...."
Response 3: The algorithms in section 3.4 are now designated as Algorithm 1 and Algorithm 2. The following sentence has been added to the previous text: "The logic of system monitoring and input processing is further detailed in Algorithm 1 and Algorithm 2, respectively"
Suggestion 4: Section 4.2: this is not a comparative analysis. Please move this under Section 4.1 since comparative analysis is still missing in the manuscript and this needs to be discussed under limitations of the study.
Response 4: The content from the previous section 4.2 was moved under section 4.1 as part of the Limitations of the study. The introductory sentence was rephrased to emphasize that this is a discussion of the limitations, not a comprehensive comparative analysis.
Suggestion 5: Section 3.5, Security and Interoperability: This looks like a future work. Please remove or move it to the future works section as this is not conducted in the current version.
Response 5: The former section 3.5 Security and Operational Compatibility has been removed from the methodology and reworked in the future tense. It is now included in section 5 Conclusion and Future Work, with a clear indication that these are planned activities.
Reviewer 2 Report
Comments and Suggestions for Authors
The authors have addressed the comments. It can be accepted.
Author Response
Thank you for your suggestion.